# The effects of set volume during isolated lumbar extension resistance training in recreationally trained males

James Steele, Adam Fitzpatrick, Stewart Bruce-Low and James Fisher

Centre for Health, Exercise and Sports Science, Southampton Solent University, Southampton, Hampshire, UK

## ABSTRACT

**Background/Aim.** Exercise designed to condition the lumbar extensor musculature is often included in resistance training (RT) programs. It is suggested that deconditioning of this musculature may be linked to low back pain. Thus effective means of conditioning these muscles are of interest to pursue. Evidence suggests that isolated lumbar extension (ILEX) RT might be most efficacious, however, the best means of manipulating resistance training variables in this regard is unknown. Set volume is an easily manipulated RT variable and one thought to also interact with training status. As such this study sought to examine the effect of set volume during ILEX RT in trained males.

**Methods.** Trained males previously engaged in exercises designed to condition the lumbar extensors underwent a 6 week intervention and were randomised to either a single set (1ST, $n = 9$), multiple set (3ST, $n = 8$) or control group (CON, $n = 9$). Pre- and post-testing of ILEX strength was conducted.

**Results.** Both 1ST and 3ST significantly increased ILEX strength ($p < 0.05$) with large effect sizes ($d = 0.89$ and $0.95$ respectively) whereas the CON group produced significant losses ($-8.9\%$) with a moderate effect size ($d = -0.53$). There was no statistically significant difference in ILEX strength gains when 1ST and 3ST were directly compared ($p = 0.336$).

**Conclusions.** The results of this study suggest that significant ILEX strength changes occur in trained males as a result of 6 weeks of ILEX RT and that these changes are unaffected by set volume.

## INTRODUCTION

Low back pain (LBP) remains a highly prevalent condition in both general (*World Health Organisation, 1998*; *Office for National Statistics, 2000*; *Office for National Statistics, 2010*; *Waddell & Burton, 2000*; *Walker, 2000*; *National Institute for Health and Clinical Excellence, 2009*) and athletic populations (*Graned & Morelli, 1988*; *Sward, Hellstrom & Jacobsson, 1990*; *Kraft, 2002*; *Bono, 2004*; *Bahr et al., 2004*; *Quinney et al., 1997*; *Haykowksy, Warburton & Quinney , 1999*). The use of resistance training (RT) to reduce injury risk has been suggested (*Stone, 1990*) and a systematic review has recently reported it to reduce sporting

Corresponding author
James Steele,
james.steele@solent.ac.uk

injury risk by one third (*Lauersen, Bertelsen & Andersen, 2013*). As evidence seems to suggest that deconditioning of the lumbar extensor musculature (lumbar erector spinae, multifidus, and quadratus lumborum) may be a factor associated with LBP and increased injury risk (*Steele, Bruce-Low & Smith, 2014a*), research has attempted to evaluate the best approaches for conditioning this musculature through RT.

*Mayer, Mooney & Dagenais (2008)* have suggested differing exercise approaches to condition the lumbar extensors which have recently been reviewed with evidence suggesting that, due to the restraint system preventing rotation of the pelvis, isolated lumbar extension (ILEX) exercise devices may be optimal (*Steele, Bruce-Low & Smith, 2013*). However, when considering an RT programme, variables such as load, effort, repetitions, repetition duration, volume, frequency, etc. should also be considered (*American College of Sports Medicine, 2009*; *Fisher et al., 2011*). The manipulation of such variables when employing ILEX has been reviewed when considering patients suffering with chronic LBP for outcomes such as pain and disability (*Steele, Bruce-Low & Smith, 2014b*), but does not exist for asymptomatic people who may be interested in injury prevention.

However, a number of studies have considered asymptomatic individuals, and the effect of RT variable manipulation upon ILEX strength as an outcome examining the volume, frequency, specificity and range of motion during training, considering both previously untrained participants (*Graves et al., 1990a*; *Carpenter et al., 1991*; *Graves et al., 1992*; *Graves et al., 1991*) as well as people with RT experience (*Tucci et al., 1992*). To date there appears only one study (*Graves et al., 1991*) which has considered training volume using ILEX but again this was on untrained participants. This research considered participants performing either one or two sets of dynamic and/or isometric ILEX training reporting no differences (*Graves et al., 1991*). Set volume in RT has historically been a contentious issue with recommendations supporting multiple set protocols (*American College of Sports Medicine, 2009*) being supported by meta-analyses (*Rhea et al., 2003*; *Peterson, Rhea & Alvar, 2004*; *Wolfe, LeMura & Cole, 2004*). Others have however critiqued these meta-analyses (*Winett, 2004*; *Otto & Carpinelli, 2006*). As the body of literature has progressed, further reviews and meta-analyses have offered support, for multiple (*Krieger, 2009*; *Frohlich, Emrich & Schmidtbleicher, 2010*), or single set protocols (*Fisher et al., 2011*; *Carpinelli, 2012*) for strength. More recently, further empirical research has examined set volume within RT with some support for multiple set approaches (*Marshall, McEwen & Robbins, 2011*; *Radaelli et al., 2014b*), and other studies finding no differences between single and multiple set routines (*Radaelli et al., 2014c*; *Radaelli et al., 2013a*; *Radaelli et al., 2013b*; *Kadir et al., 2014*; *Adnan et al., 2014*; *Correa et al., 2014*; *Baker et al., 2013*).

Differentiating between trained and previously untrained participants within the research appears to be an important consideration as it is suggested that the level of trainability affects the interaction of specific RT variable manipulation with outcomes, particularly with respect to set volume (*Frohlich, Emrich & Schmidtbleicher, 2010*). In addition, that trained people should perform a larger training volume than untrained people (*American College of Sports Medicine, 2009*). Though *Graves et al. (1991)* reported no difference between one and two sets of ILEX, the authors only considered untrained

participants and did not consider the effects of greater set volumes typically suggested as being optimal for strength (≥3 sets; *Krieger, 2009*). Since RT is shown to be an effective intervention in trained and sporting populations for reducing injury risk, it is of interest to investigate the effects of set volume during ILEX exercise in a trained population looking specifically at the lumbar extensor muscles. As such, the aim of the present study was to consider the effects of single and multiple set ILEX exercise in trained males.

## MATERIALS & METHODS

### Study design

A randomised controlled trial design was adopted with three experimental groups examining the effects of set volume upon ILEX strength in recreationally trained participants. The study design was approved in writing by the relevant ethics committee at the author's institution.

### Participants

Participants were required to be males between 18 and 30 years old, have been involved in a RT programme for a minimum of 6 months at a minimum frequency of 2x/week including participation in exercises designed to condition the lumbar extensors (*Mayer, Mooney & Dagenais, 2008*; *Steele, Bruce-Low & Smith, 2013*) at the beginning of the study. Furthermore, participants were to have no medical condition for which RT is contraindicated to participate. Specific exclusion criteria included: (1) history of LBP or lumbar spine pathologies or known deformities; (2) currently experiencing LBP; (3) knee or hip disorders contraindicating use of the ILEX device; (4) inability or unwillingness to cease participation in other lumbar conditioning exercises for the duration of the study; and (5) currently taking any illegal ergogenic aids or nutritional supplementation. Written informed consent was obtained from all participants.

Power analysis of research using ILEX (*Graves et al., 1990a*) was conducted to determine participant numbers ($n$) using an effect size (ES), calculated using Cohen's $d$ (*Cohen, 1992*), of 1.26 for improvements in ILEX strength. Participant numbers were calculated using equations from *Whitley & Ball (2002)* revealing each group required 10 participants to meet required power of $\beta = 0.8$ at an alpha value of $p \leq 0.05$.

Thirty participants were initially identified and recruited. Figure 1 shows a CONSORT diagram highlighting participant numbers for enrolment, allocation, follow-up and analysis stages. No initial dropouts were recorded after eligibility assessment so 30 participants were randomised using a random drawing of cards to one of three groups; a single set group (1ST; $n = 10$), a multiple set group (3ST; $n = 10$), and a non-training control group (CON; $n = 10$).

### Equipment

Stature was measured using a stadiometer (Holtan ltd, Crymych, Dyfed, Wales), body mass measured using scales (SECA, Hamburg, Germany) and Body Mass Index (BMI) calculated. Isometric ILEX strength testing, range of motion and training were performed using an ILEX device (MedX, Ocala, Florida, USA). The device has been shown reliable in

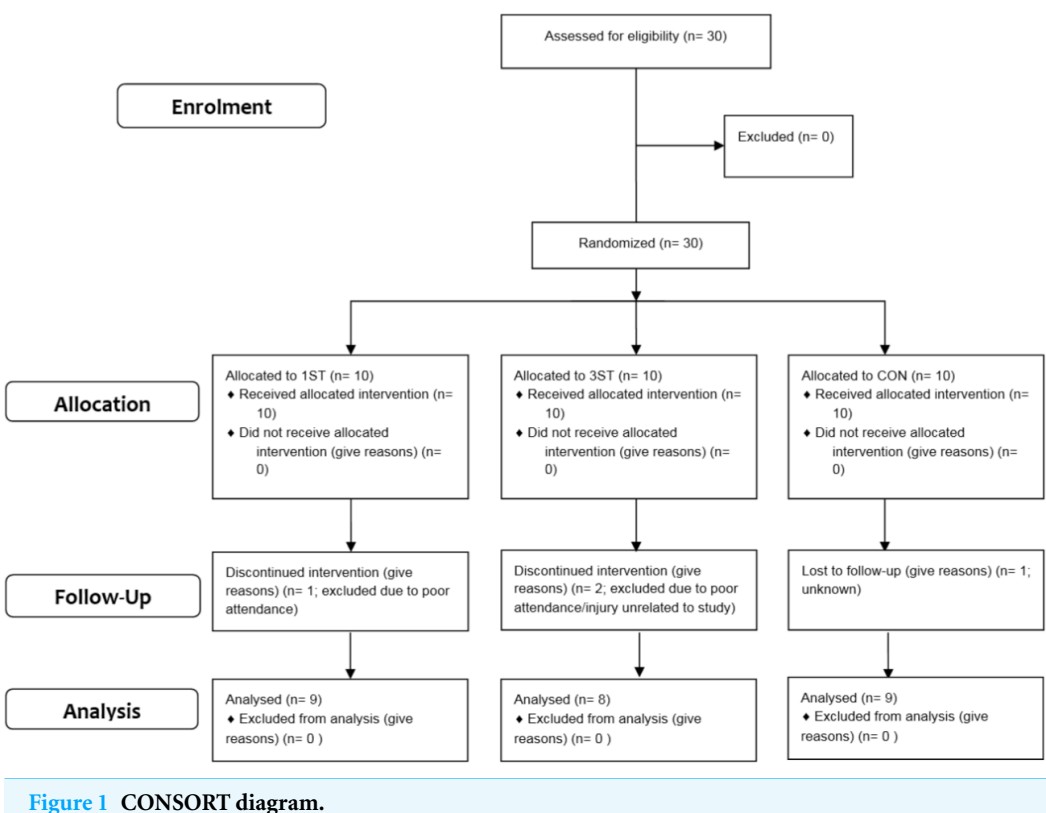

**Figure 1 CONSORT diagram.**

assessing isometric strength at repeated angles in asymptomatic ($r = 0.81–0.97$; *Graves et al., 1990b*) and symptomatic participants ($r = 0.57–0.93$; *Robinson et al., 1992*), and valid in measurement (*Pollock et al., 1991*; *Inanami, 1991*).

## Participant testing

ILEX strength was tested twice, on separate days (at least 72 h apart to avoid residual fatigue or soreness) before with the first acting as a familiarisation, and once after the RT intervention. Each test using the ILEX device involved maximal voluntary isometric contractions at various angles through the participant's full range of motion. Details of the full test protocol using the MedX and its restraint mechanisms have been documented elsewhere (*Graves et al., 1990b*). ILEX strength averaged across the participant's full range of motion was considered.

## Participant training

Training was conducted at a frequency of 1x/week for a period of 6 weeks. This frequency of training has been shown to significantly improve ILEX strength and was chosen over more frequent training (*Bruce-Low et al., 2012*) due to potential for overtraining when the lumbar extensor muscles are isolated (*Graves et al., 1990a*). Both groups performed one set of variable resistance ILEX exercise. Both performed a warm-up set lasting 60 s using 50% of their training load. Starting training load was 50% of max recorded tested functional torque (TFT) during maximal isometric testing for both groups and repetitions performed

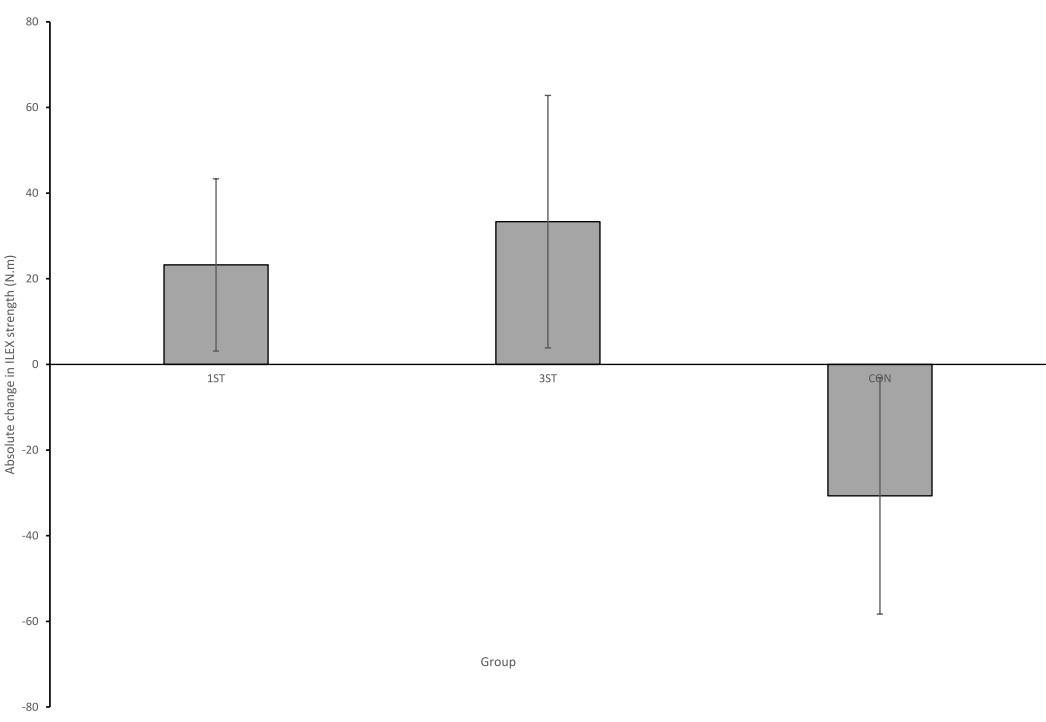

**Figure 2** **Mean change in ILEX strength with 95% CIs.**

until momentary muscular failure (MMF) in order to control for effort (*Steele, 2014*). Repetitions were performed taking at least 2 s to complete the concentric phase, holding for 1 s in full extension and taking at least 4 s for the eccentric phase. The 3ST group rested for 3 min between each set during which time participants remained seated in the ILEX device with the restraints loosened. Resistance load was increased by 10% in the next session or set once the participant was able to perform more than 12 repetitions using their current load before achieving MMF. However, in order to maintain a similar repetition range for each set in the 3ST group, load was reduced by 5% in the next set if the participant was unable to complete at least 8 repetitions before achieving MMF (*Medeiros Jr et al., 2013*). All groups were asked to refrain from using any other lumbar conditioning exercises for the duration of the study.

## Data analysis

Participants that missed 2 or more sessions were excluded from analysis. Twenty six participants' (1ST $n = 9$; 3ST $n = 8$; CON $n = 9$) data were available for analysis. Assumptions of normality of distribution were examined using a Kolomogorov–Smirnov test. As not all data sets met assumptions of normality of distribution, non-parametric analyses were performed. Demographic characteristics, baseline ILEX strength and the effects of the group (independent variable) upon the absolute change in ILEX strength (dependent variable) was examined using a Kruskal–Wallis one way analysis of variance. Significant results from the Kruskal–Wallis test were further subjected to post hoc planned comparisons using Mann–Whitney U tests between 1ST and 3ST. Further, 95% confidence

**Table 1 Participant demographic characteristics.**

|  | 1ST | 3ST | CON |
|---|---|---|---|
| Participants (No.) | 9 | 8 | 9 |
| Age (years) | 21 ± 2 | 21 ± 1 | 20 ± 1 |
| Stature (cm) | 174.7 ± 7.6 | 178.5 ± 8.7 | 178.4 ± 6.3 |
| Body mass (kg) | 74.3 ± 9.1 | 78.7 ± 10.3 | 77.7 ± 7.8 |
| Training experience (years) | 3 ± 1 | 5 ± 2 | 3 ± 1 |
| Baseline ILEX strength (Nm) | 277.66 ± 99.99 | 335.71 ± 80.04 | 339.84 ± 64.82 |
| ILEX range of motion (°) | 72 | 72 | 72 |

intervals (CI) were calculated in addition to ES using Cohen's $d$ (*Cohen, 1992*) for each outcome to examine significance and magnitude of effects within groups where significant changes were considered where 95% CIs did not cross zero and for magnitudes an ES of 0.20–0.49 was considered as small, 0.50–0.79 as moderate and ≥0.80 as large. Statistical analysis was performed using SPSS (IBM Statistics for Windows, Version 20.0; IBM, Portsmouth, Hampshire, UK) and $p \leq 0.05$ set as the limit for statistical significance.

## RESULTS & DISCUSSION

### Participant demographics

Participant demographics are shown in Table 1. There were no significant between group effects for any demographic characteristics.

### Strength

Figure 2 shows mean changes and 95% CIs for ILEX strength for each group. The Kruskal–Wallis one way analysis of variance revealed a significant between groups effect for ILEX strength ($X^2(2) = 11.645, p = 0.003$; mean rank for 1ST = 16.22, 3ST = 18.25, CON = 6.56). Planned comparison between 1ST and 3ST using Mann–Whitney U revealed no significant difference for ILEX strength ($21.00U = 26.000, p = 0.336$; Median ± Interquartile Range, 28.48 ± 43.49 and 46.30 ± 40.58 respectively). The 95% CIs indicated that both 1ST and 3ST groups significantly increased strength (8.31% and 10.68% respectively) with ESs considered large (0.89 and 0.95 respectively) whereas there was a significant reduction in ILEX strength in the control group (−8.9%) with a moderate ES (−0.53).

## DISCUSSION

This study examined the effects of set volume in recreationally trained male participants performing an ILEX RT intervention. The results indicated that both 1ST and 3ST significantly improved their ILEX strength with similarly large ESs, whereas the CON group lost strength over the intervention period. No differences between 1ST and 3ST for ILEX strength change were found as a result of planned comparisons.

Reviews and meta-analyses offer divergent conclusions regarding the effects of set volume (*Fisher et al., 2011*; *Otto & Carpinelli, 2006*; *Rhea et al., 2003*; *Peterson, Rhea & Alvar, 2004*; *Wolfe, LeMura & Cole, 2004*; *Winett, 2004*; *Krieger, 2009*; *Frohlich, Emrich*

& *Schmidtbleicher, 2010*; *Carpinelli, 2012*). Within this study, and in contrast to earlier research by *Graves et al. (1991)* examining set volume and ILEX RT, we used trained participants, as it is believed that training status might interact with the effects of set volume (*Frohlich, Emrich & Schmidtbleicher, 2010*). In particular, it is recommended that greater set volumes be utilised by trained people (*American College of Sports Medicine, 2009*). The results of this study seem to suggest that, at least for ILEX training, set volume does not impact upon strength changes in trained people. This finding is in agreement with the majority of more recent empirical research regarding set volume indicating multiple sets appear to offer similar efficacy upon strength gains to single set approaches both in untrained (*Radaelli et al., 2014b*; *Radaelli et al., 2014c*; *Radaelli et al., 2013a*; *Radaelli et al., 2013b*; *Kadir et al., 2014*) and recreationally trained participants (*Adnan et al., 2014*; *Baker et al., 2013*). However, research by *Marshall, McEwen & Robbins (2011)* in trained males does contrast this. Though they reported no difference between one and four sets, they noted significantly greater strength changes with eight sets. Though this may have been due to this group containing a high proportion of medium/high responders (*Carpinelli, 2012*), it remains a possibility that a higher set volume than that used in the present study ($\geq 3$ sets) may induce greater ILEX strength gains. Indeed, another recent study by *Radaelli et al. (2014a)* suggests a dose response relationship between 1, 3 and 5 sets in untrained yet active males. However, it should be noted that the lumbar extensors have been suggested to be particularly prone to overtraining (*Graves et al., 1990a*). All participants within the 3ST group reported feeling nauseous and entirely exhausted (not limited to the lumbar extensors) during training, whereas only one of the 1ST group displayed these symptoms. *Hass et al. (2000)* reported that dropout rate was higher for a multiple set RT program compared with one employing single set. Twenty-five percent dropped out from the multiple set group (5 for lack of adherence and 2 for injuries) compared with none in the single set group. Though no injuries were reported from our participants due to the training, the potential of increased injury risk and impact of higher volume high effort training upon adherence, in addition to the increased time required, should be weighed against any potential for greater gains with much higher set volumes.

It is worth discussing that ILEX may be unique even when considering trained people. As noted, the participants in this study were previously participating in RT involving exercises suggested by *Mayer, Mooney & Dagenais (2008)* to condition the lumbar extensors (including deadlifts, good mornings, trunk extension machines etc.). However, though two participants had previously used the ILEX device used in this study (though the specific duration or implementation of this was not known), not all participants had been previously involved in ILEX training specifically. *Pollock et al. (1989)* also reported considerable improvements in average ILEX strength ($\sim$37%) in trained individuals (at least 1 year training experience) engaged in similar exercises prior to ILEX RT interventions (including use of a commercially available trunk extension resistance machine). *Pollock et al. (1989)* speculated that, although these participants had been engaged in exercises intended to condition the lumbar extensors, their lumbar extensors may have in fact been untrained relative to their other musculature. The large ESs in

**Peer**J ___________________________________________________________

our trained participants further suggests this may be the case. *Steele, Bruce-Low & Smith (2013)* concluded in their review of the specificity of exercises designed to condition the lumbar extensors that many exercises suggested by *Mayer, Mooney & Dagenais (2008)* may be inferior to ILEX. Indeed, it may be that the lumbar extensor musculature may become deconditioned relative to other musculature due to the lack of conditioning effect from many typical trunk extension based exercises or movements (*Steele, Bruce-Low & Smith, 2014a*). Our CON group in fact lost ILEX strength over the intervention period highlighting this possibility (though it is unclear as to why a decrease of comparable magnitude to both the 1ST and 3ST groups' increases was found). Thus, it remains possible that set volume may impact upon strength gains in those previously engaged in ILEX RT who thus have trained lumbar extensors. This is an area that requires further investigation.

The practical implications of these findings are of importance. The typical recommendations for a single set approach for ILEX has been noted as time efficient for addressing pain and disability in people with chronic LBP (*Steele, Bruce-Low & Smith, 2014b*). However, as noted, prior reviews have not offered recommendation as to what set volume might be best for ILEX RT for the purposes of conditioning the lumbar extensors (*Steele, Bruce-Low & Smith, 2013*), something of particular important for those wishing to reduce risk of low back injury (*Stone, 1990*; *Lauersen, Bertelsen & Andersen, 2013*; *Steele, Bruce-Low & Smith, 2014a*; *Steele, Bruce-Low & Smith, 2014b*). This study suggests that a single set approach may be a time efficient method to effectively condition the lumbar extensors. Including the warm-up set, the single set group performed between 1.56 and 2.24 min of exercise, whereas the multiple set group, including between set rest periods, trained for between 9.48 and 11.12 min from beginning exercise to end. Though this may appear a practically small difference, in the context of a training program including a selection of other exercises that are performed using either single or multiple sets this could add up to considerably more training time.

Limitations of the present study should be considered. Though our sample size was similar to other research examining single and multiple set RT, due to drop outs, we fell slightly below the sample size indicated by our power analysis (10 participants). However, calculations of observed power for group effects upon ILEX strength changes were $\beta = 0.966$, and thus this study can be considered as adequately powered to detect between group differences. Despite the study being adequately powered, the limitations to practical conclusions should be noted. The study was only 6 weeks in duration and thus, although other studies of longer duration have suggested no difference, it remains the possibility that differences between single and multiple set ILEX in trained participants might manifest over longer training durations. Further, recent studies considering the effects of set volume have included greater than 3 sets and suggested that both 5 (*Radaelli et al., 2014a*) and 8 sets (*Marshall, McEwen & Robbins, 2011*) may produce greater strength adaptations. Whether higher set volumes may produce greater increases for ILEX RT remains to be investigated.

## CONCLUSION

To conclude, the results of this study suggest that strength changes resulting from an ILEX RT intervention in recreationally trained males are unaffected by differing set volumes. We compared one and three set training finding no difference in the magnitude of strength changes that occurred. Both approaches appeared to significantly improve ILEX strength. Thus, we can recommend that a single set approach to ILEX RT is a time efficient and sufficient approach for the purposes of conditioning the lumbar extensors. Indeed, considering evidence that RT can be used for reducing injury risk, this might also represent a time efficient approach for primary prevention of low back injury and pain, and future research should look to investigate this potential.

### Funding
The authors declare there was no funding for this work

### Competing Interests
The authors declare there are no competing interests.

### Author Contributions
- James Steele analyzed the data, wrote the paper, prepared figures and/or tables, reviewed drafts of the paper.
- Adam Fitzpatrick and Stewart Bruce-Low conceived and designed the experiments, performed the experiments, reviewed drafts of the paper.
- James Fisher reviewed drafts of the paper.

### Human Ethics
The following information was supplied relating to ethical approvals (i.e., approving body and any reference numbers):

Southampton Solent University Health, Exercise and Sport Science Ethics Committee provided written approval.

### Supplemental Information
Supplemental information for this article can be found online at http://dx.doi.org/10.7717/peerj.878#supplemental-information.

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
