# Peer review of "The effects of set volume during isolated lumbar extension resistance training in recreationally trained males"

_PeerJ, doi:10.7717/peerj.878_

## Round 0.1 · original submission · Minor Revisions

Dear Prof. Steele,

Both reviewers pointed out that your paper contained interesting material that deserves publication. I find both reviews informative and helpful and I ask you to consider the suggestions carefully and in all neccessary detail.

·

Basic reporting

The writing style, paragraph and sentence structure, and grammar were well done.

The authors have done an excellent job in reporting their data analysis with confidence limits, effect sizes and exact p-values (even for results that were not significant).

I’m not comfortable that the first two sentences of the abstract being about LBP. The same could be said about the manuscript’s introduction. This manuscript is not about LBP nor was there any measure relating to LBP. This manuscript is about training volume. That the muscles trained may have implications in LBP may feature late in the discussion, but making LBP such a prominent point is misleading the reader from the start about LBP when, in fact, no such conclusion can be drawn from the data.

Experimental design

I was surprised at a protocol that takes its participants to failure on back extensions. Can the authors please provide some comment on the risk of lower back injury by fatiguing supporting muscles during a flexion/extension protocol? Is injury why three of the four participants withdrawing from the study were from training groups?

Otherwise, the use of the randomized cross-over design is appropriate to address the research question.

Validity of the findings

Data analysis – as a suggestion, the authors may be able to log transform their data to fix their distribution problems. Using non-parametric tests typically return lower p-values, thus (potentially artificially) tightening the confidence limits. If the distribution problem could be fixed by running a parametric statistic on log transformed data, then that would be better (albeit, transforming does not always help fix distribution problems).

Otherwise, I emphatically support and commend the authors’ use of 95% confidence limits and effect sizes.

Additional comments

Line 31 – there is no purpose to including “respectively”. Please delete it.

Line 66, 67, 180, etc. – “persons” should be used only when there is a defined number, but “people” should be used for an undefined group (“63 persons” compared to “trained people”).

Line 148 – need more detail on the manufacturer’s information of SPSS (i.e. IBM SPSS Statistics for Windows, Version 20.0. Armonk, NY: IBM Corp.)

·

Basic reporting

Line 34 – There is one more muscle to cite, square lumbar. According to McGil et al. (1996) it is the only muscle highly activated during extension, flexion and lateral flexion of the spine. Miranda (2000) states that the bilateral contraction of this muscle produces the trunk extension.
Line 66 - something seems to be missing in this quote

Experimental design

Line 82 - Since the participants were currently practicing exercises to condition the lumbar extensors, it would be interesting to report their specific history regarding set volume. Were they training with one or three sets? Which exercise they were performing?...
Line 111 - Did you calculate the r for the repeated tests? There was any procedure if the values were divergent (i.e. repeat the test if the difference between the first and the second was > 5%)

Validity of the findings

Line 163 - the ES found here seems to be pretty large for a 6 week intervention trained subjects, this reinforces the need for detailing their training history
Line 178 – There are others references that affirm the opposite, that trained people should train with less volume when compared to untrained people. In the meta-analysis of Rhea et al (2003), for example, the number of sets were the same for trained and untrained people; however the training frequency was lower for trained people, leading to a low weekly volume.
Line 193 - Maybe you can add the high dropout rate found by Haas et al. (2000) when the number of sets was increased from one to three in trained people.
Line 248 – It should be said that besides the lumbar conditioning improvement, this strengthening suggests a prevention to low back pain disorders. This conclusion could match with the beginning of the text when the author refers to high incidence in population of low back pain.

Additional comments

This is a very interesting paper that analyzed the effects of number of sets in lumbar extensors muscle strength of trained individuals. The information brought by the study is extremely relevant, because it allow the design of time efficient programs for strengthening the lumbar extensors, which may have many important applications for low back pain prevention and management. Studying trained persons is one of the strengths of the manuscript, since literature regarding this population is scarce. I would like to make a few observations and kindly ask the author to consider them, as I think they would improve the manuscript.

---

## Round 0.2 · accepted · Accept

I've checked the paper and am pleased. I think that this is a valuable contribution to the field.